# Prevalence and Genotypes of Cryptosporidium in Wildlife Populations Co-Located in a Protected Watershed in the Pacific Northwest, 2013 to 2016

**DOI:** 10.3390/microorganisms8060914

**Published:** 2020-06-17

**Authors:** Xunde Li, Tran Nguyen, Chengling Xiao, Ann Levy, Yone Akagi, Sarah Silkie, Edward R. Atwill

**Affiliations:** 1Western Institute for Food Safety and Security, University of California, Davis, CA 95616, USA; xdli@ucdavis.edu (X.L.); tphnguyen@ucdavis.edu (T.N.); clxiao@UCDAVIS.EDU (C.X.); 2Department of Population Health and Reproduction, School of Veterinary Medicine, University of California, Davis, CA 95616, USA; 3Portland Water Bureau, Portland, OR 97227, USA; Ann.Levy@portlandoregon.gov (A.L.); Yone.Akagi@portlandoregon.gov (Y.A.); Sadie.Silkie@portlandoregon.gov (S.S.)

**Keywords:** *Cryptosporidium*, species, genotypes, wildlife, protected watershed, water supply

## Abstract

Between October 2013 and May 2016, 506 scat samples were collected from 22 species of wildlife located in a protected watershed of a major municipal water supply in the Pacific Northwest, USA. Overall prevalence of *Cryptosporidium* in the wildlife scat was 13.8% (70/506), with 15 species of wildlife found positive for *Cryptosporidium*. Prevalence of *Cryptosporidium* varied among species of wildlife, with higher prevalences observed in cougars (50.0%), mountain beavers (40.0%), and bobcats (33.3%), but none of these species are riparian-dependent. Genotyping of *Cryptosporidium* by sequencing PCR amplicons from the 18S rRNA gene were successful for seven species of wildlife, including bobcat, unknown predator, black-tailed deer, deer mouse, snowshoe hare, mountain beaver, and western spotted skunk. BLAST and phylogenetic analyses indicated that multiple species and genotypes of *Cryptosporidium* were present, with some isolates possibly co-circulating within and between wildlife populations in this protected watershed. Evidence of oocyst exchange between infected prey and their predators was also found. During the study period, several zoonotic *Cryptosporidium* species and genotypes that are uncommon in humans were detected in bobcat (99.58% identical to *Cryptosporidium felis*), unknown predator (100% identical to *Cryptosporidium canis*), snowshoe hare (100% identical to *Cryptosporidium* sp. skunk genotype), and mountain beaver (100% identical to *Cryptosporidium ubiquitum*). Novel sequences were also found in mountain beaver. To our knowledge, this is the first published report of a unique genotype or species of *Cryptosporidium* in mountain beaver (*Aplodontia rufa*).

## 1. Introduction

Parasites of the genus of *Cryptosporidium* infect virtually all vertebrate groups, with some species exhibiting host specificity [1,2]. Currently, approximately 31 valid species of *Cryptosporidium* have been described [3,4,5]. For example, three species infect fish (*Cryptosporidium molnari*, *Cryptosporidium huwi*, and *Cryptosporidium scophthalmi*); one species infects amphibians (*Cryptosporidium fragile*); two species infect reptiles (*Cryptosporidium serpentis* and *Cryptosporidium varanii*); four species infect birds (*Cryptosporidium meleagridis*, *Cryptosporidium baileyi*, *Cryptosporidium galli,* and *Cryptosporidium avium*); and twenty one species infect mammals (*Cryptosporidium muris*, *Cryptosporidium parvum*, *Cryptosporidium wrairi*, *C. felis*, *Cryptosporidium andersoni*, *C. canis*, *Cryptosporidium hominis*, *Cryptosporidium suis*, *Cryptosporidium bovis*, *Cryptosporidium fayeri*, *Cryptosporidium macropodum*, *Cryptosporidium ryanae*, *Cryptosporidium xiaoi*, *C. ubiquitum*, *Cryptosporidium cuniculus*, *Cryptosporidium tyzzeri*, *Cryptosporidium viatorum*, *Cryptosporidium scrofarum*, *Cryptosporidium erinacei*, *Cryptosporidium rubeyi,* and *Cryptosporidium proliferans*) [5,6,7]. *C. hominis* and *C. parvum* are responsible for the majority of human infections and are therefore of higher public health concern [8,9].

*Cryptosporidium* is one of the most common causes of waterborne zoonotic enteric disease worldwide [5]. For example, a massive waterborne outbreak of cryptosporidiosis occurred in Milwaukee, Wisconsin, in 1993, which was transmitted through the public water supply to affect over 400,000 people [10]. A similar community outbreak of cryptosporidiosis infected about 13,000 people in Georgia in 1987, also due to contamination of the public water supply [11]. On the west coast, a community outbreak of cryptosporidiosis associated with surface water-supplied municipal water occurred in Baker City, Oregon in 2013, in which 2780 people were infected [12]. Active and passive surveillance for waterborne *Cryptosporidium* spp. oocysts have detected this parasite in numerous municipal and rural watersheds around the world. For example, oocysts have been detected in the South Nation watershed in Ontario [13,14] and in British Columbia [15], Canada, in the Upper-Sûre watershed in Luxembourg [16], and coastal watersheds in France [17]. In the USA., detection of *Cryptosporidium* spp. oocysts in water have been widely reported, such as during base- and storm-flow conditions for the Potomac River watershed [18], in surface water in the Wachusett Reservoir watershed in central Massachusetts [19], during storm flow conditions from watersheds in New York [20], popular swimming sites in central California [21], and in rivers in Washington State [22].

Because sources of drinking water such as lakes, reservoirs, rivers, and groundwater can be contaminated by waterborne pathogens shed by wild and domestic animals, the prevalence and distribution of zoonotic species of *Cryptosporidium* in wildlife located within watersheds that provide source water for municipal drinking water is of public health importance [5]. The objective of this project was to determine the prevalence and diversity of known species and novel genotypes of *Cryptosporidium* in the resident and migratory wildlife populations located in a major, protected watershed in the Pacific Northwest; such data would contribute to a better assessment of *Cryptosporidium* health risks from human exposure to this parasite through drinking water. The protected watershed area is 100,000 acres and almost entirely forested, primarily within national forestland. The roads to the watershed are gated to prevent public entry. The drinking water system based on the watershed is unfiltered and does not treat for *Cryptosporidium*.

## 2. Materials and Methods

### 2.1. Wildlife Scat Samples

Between October 2013 and May 2016, naturally-voided wildlife scat samples were opportunistically collected off the ground along a network of roads, sections of trail, and sample locations in the forest throughout the watershed of a major municipal drinking water supply in the Pacific Northwest, USA. Samples were visually inspected for freshness based on a sheen indicating high moisture content, remaining fraction of digestion by-products, evidence of needles or leaves, and pliability. Additional scat samples were collected from live-trapped small mammals such as mice and voles given the small size of their scat and therefore difficulty in finding naturally voided samples in the environment. This sampling design was intended to collect scat from a wide range of resident and migratory wildlife from throughout the watershed, with the number of scat samples per wildlife species in part the result of the varying density of different free-ranging wildlife on the watershed, seasonal migration of some species, and the likelihood of each species defecating along the sampling locations. Collection of scat samples was conducted approximately two to five times per month, with snow or heavy rainfall preventing sampling. Depending on the sampling schedule and availability of naturally-voided scat or trapped small mammals, 1 to 25 fresh scat were collected per month by wildlife professionals or trained staff who determined the species of trapped wildlife or scat. Scat were identified in the field based on the diameter, length, shape, color, structure, and contents, taking into account adjacent wildlife tracks and markings. However, there were challenges in identifying scat between some species, for example, scat from predators such as coyote, cougar, and bobcat can be difficult to distinguish. Samples were placed in sterile zip bags and stored under refrigeration until shipped to UC Davis within 24 h of collection.

### 2.2. Detection of Cryptosporidium Oocysts from Scat Samples

Within 24 h of receiving the shipment, scat samples were processed for detection of *Cryptosporidium* oocysts and *Giardia* cysts. Zip bags of feces were manually massaged for ~30 s to break down large aggregates and homogenize the fecal material. After removing visible large fibers, if any, 10 g of sample were transferred to a 50 mL conical tube and diluted with 40 mL of deionized water. The tube was homogenized using a vortex and then centrifuged at 1500× *g* for 10 min. The supernatant was removed by aspiration, leaving the pellet and supernatant at a ratio of 1:1 (*v:v*). The sample was re-homogenized and 10% of the fecal suspension (equivalent to 1.0 g of the original sample) was transferred to a new tube for IMS and IFA. If the original mass of a sample was <10 g, the entire sample was processed except for reserving ~0.5 g of feces. Scat samples from small rodents like mice or voles were usually less than 0.1 g. For these samples, one third of the mass was reserved for PCR and two thirds were processed by homogenizing in PBS at a 1:10 ratio (fecal:PBS).

Depending on wildlife species and the preference indicated by the funding agency, one of four different methods would be used to process the sample: direct immunofluorescence assay (IFA), immunomagnetic separation (IMS) coupled with immunofluorescence assay (IMS-IFA), pre-treatment with sodium pyrophosphate (NaPP) solution followed by IMS coupled with immunofluorescence assay (NaPP-IMS-IFA), or pre-treatment with diethyl ether solution followed by IMS coupled with immunofluorescence assay (Ether-IMS-IFA). The rationale for using different analytical methods was because different wildlife species have different diets, resulting in different levels of fat or other constituents in the scat that needed special processing for detecting oocysts. The IMS procedure was performed by using an automatic BeadRetriever (Invitrogen, Carlsbad, CA, USA) using Dynabeads GC-Combo (Applied Biosystems, Foster City, CA, USA). The IFA method was implemented as described previously [23,24] using the Aqua-Glo G/C Direct kit (Waterborne Inc., New Orleans, LA, USA). Slides were examined using a fluorescence microscope (Olympus BX 60) at × 200–400 magnification and round or oval objects with 4 to 7 µm in diameter, green fluorescence, and appropriate morphology were considered as oocysts.

For each wildlife species, a quality control protocol was implemented by spiking scat samples with ColorSeed that contained 100 oocysts (BTF, Biomerieux, Sydney, Australia) at a standard frequency of one quality control scat sample out of every ten scat samples that were analyzed. The ColorSeed oocysts were spiked into the processed samples mentioned above by adding the oocyst suspension directly into the 50 mL conical tubes containing the processed samples, followed by washing the ColorSeed tube two times with 0.05 % Tween 20 and the rinsate also added to the sample tube. Spiked samples were then processed with the same analytical procedures as described above and percent recoveries were determined by enumeration of the spiked 100 ColorSeed™ oocysts.

### 2.3. PCR and Sequencing

Approximately 0.2 mL of retained fecal suspension or the reserved small rodent feces from samples that were microscopic positive for *Cryptosporidium* oocysts were subjected to 5 cycles of freeze (−80 °C) and thaw (+70 °C) and used for DNA extraction using the QIAamp DNA Stool Mini Kit (Qiagen Inc., Valencia, CA, USA) according to the manufacturer′s instructions. Amplification of a fragment of the 18S rRNA gene by nested-PCR was performed using primers and cycling conditions as previously described [20,25,26]. AmpliTaq DNA polymerase (Thermo Fisher Scientific, Grand Island, NY, USA) was used for all PCR amplifications. A positive control using DNA of *C. parvum* isolated from infected calves from a dairy near Davis, CA, as template and a negative control without DNA template were included in each round of PCR. PCR products were verified by electrophoresis in 2% agarose gel stained with ethidium bromide. Products of the secondary PCR were purified using the QIAamp DNA Mini Kit (Qiagen Inc.) according to the manufacturer′s instructions. Purified DNA was sequenced in both directions at the University of California DNA Sequencing Facility using an ABI 3730 Capillary Electrophoresis Genetic Analyzer (Applied Biosystems Inc., Foster City, CA, USA).

### 2.4. BLAST and Phylogenetic Analysis

Sequences were analyzed and consensus sequences were generated using the Vector NTI Advanced 11 software (Invitrogen). Consensus sequences were compared to *Cryptosporidium* sequences in the GenBank using NCBI′s online BLAST tool with the default algorithm parameters to target 100 sequences (http://blast.ncbi.nlm.nih.gov/) (12 March, 2020, as last day accessed). A phylogenetic tree was constructed using the Vector NTI Advance 11 based on a pairwise alignment. Depending on the availability of the 18S rRNA gene sequences of *Cryptosporidium* in the GenBank, reference sequences for constructing the phylogenetic trees were selected based on: (1) sequences representing well described *Cryptosporidium* species from fish, amphibians, reptiles, birds, and mammals; (2) sequences of known zoonotic genotypes; (3) sequences previously used by other investigators for species description or as reference sequences; (4) sequence length (longer sequence available for each species; i.e., ≥700 bp); and (5) sequences not originating from cloned PCR products due to the potential for erroneous sequence data generated from cloning PCR products [27,28]. Names and GenBank accession numbers of selected references sequences are shown in Figure 1.

### 2.5. Statistical Analysis of Apparent Prevalence

Stata Intercooled statistical software (Version 14, College Station, TX, USA) was used to test whether the prevalence ratio (PR) for comparing different orders of mammals was significantly different from 0 and to calculate the 95% confidence interval. As an example, the order Rodentia with a scat prevalence of 8% (14/174) is the denominator and the numerator is the 21.2% prevalence from the order Carnivora. In this example the PR of Carnivora versus Rodentia is 2.64, calculated as (oocyst prevalence from Carnivora/oocyst prevalence from Rodentia) = (31/146)/(14/174) = (0.212/0.0804) = 2.64.

## 3. Results

### 3.1. Matrix Spike Recovery

Matrix spike recovery trials were performed on scat from 11 wildlife species during the study period. Variable percent recoveries were observed in scat from different wildlife species, with the highest (31.5%) in cougar and lowest (2.2%) in river otter samples (Table 1). This suggests that for scat samples with low numbers of oocysts/g feces, there may be some false negative assay results for scat from wildlife species with low matrix spike recoveries.

### 3.2. Apparent Prevalence of Cryptosporidium in the Wildlife Populations

In total 506 scat samples were collected from 22 species of resident or migratory wildlife from throughout the watershed between October 2013 and May 2016. Overall, 13.8% (70/506) of the samples were positive for *Cryptosporidium* oocysts. Sixty-eight percent (15/22) of the wildlife species sampled had one or more scat test positive for *Cryptosporidium* oocysts during the study period (Table 2). Prevalence of *Cryptosporidium* varied widely between wildlife species, with a higher prevalence detected in cougar (4/8 or 50.0%), mountain beaver (4/10 or 40%), and bobcat (15/45 or 33.3%), excluding the high percentages in species due to smaller sample sizes like pika (1/1 positive) and western spotted skunk (1/2 positive) (Table 2). The prevalence values reported above are for the apparent prevalence, which is unadjusted for the sensitivity and specificity of the diagnostic assay. Assays with sensitivity less than 100% can result in false negatives and decrease the apparent prevalence.

At the level of order, with Rodentia functioning as the referent or denominator for calculating the prevalence ratio (PR), the prevalence of *Cryptosporidium* in scat was significantly greater for wildlife grouped under Carnivora (PR = 2.64, *p* = 0.0007, 95% CI 1.46–4.77) and Artiodactyla (even-toed ungulates) (PR = 2.39, *p* = 0.007, 95% CI 1.25–4.55), but not significantly different (*p* > 0.05) from Eulipotyphla (shrews), Lagomorpha (hares, rabbits, and pikas), or Anseriformes (only geese were involved in this project from this order). Alternatively, with Lagomorpha functioning as the referent or denominator for calculating the PR, the prevalence of *Cryptosporidium* in scat was marginally greater for wildlife grouped under Carnivora (PR = 2.25, *p* = 0.056, 95% CI 0.92–5.48), but not significantly different (*p* > 0.05) from Artiodactyla, Eulipotyphla, or Anseriformes. All other comparisons of the prevalence by order were not significantly different (*p* > 0.05).

### 3.3. Genotypes of Cryptosporidium in the Wildlife Populations

Genotyping isolates of *Cryptosporidium* oocysts by PCR and sequencing a fragment of the 18S rRNA gene were successful for 12 scat samples from 7 species of wildlife, including bobcat (2), unknown predator (1), black-tailed deer (1), deer mouse (3), snowshoe hare (1), mountain beaver (3), and western spotted skunk (1) (Table 3). GenBank accession numbers of these sequences are MT524964–77.

Comparisons of the 18S rRNA gene sequences of these isolates to reference sequences of different species and unique genotypes *Cryptosporidium* in GenBank are shown in Table 3. One isolate (scat #1017) from a bobcat (*Lynx rufus*) was highly identical (99.57 to 99.58%) to two isolates of *C. felis* (MK886594 and MF589920) from domestic cats (*Felis catus*) and 99.47% identical to an isolate of *C. felis* (HM485433) from humans. The other isolate from a bobcat (scat #1108) was only 97.32 to 97.96% identical to *Cryptosporidium* isolates from storm water (unknown host) and 97.23% identical to an isolate from a meadow vole (*Microtus pennsylvanicus*), which suggests that this isolate is a novel genotype in bobcats. The isolate from black-tailed deer (*Odocoileus hemionus columbianus*) (scat #930) was 100% identical to thirteen isolates of *Cryptosporidium* sp. deer genotype from different species of deer. In addition, this black-tailed deer isolate was also 100% identical to an isolate of *C. ryanae* (MK982509) from barking deer (*Muntiacus muntjak*). Among the three isolates from deer mice (*Peromyscus maniculatus*), two isolates (scat #R5 and #R6) were 100% identical to an isolate of *Cryptosporidium* sp. deer mouse genotype (KX082685) and the other isolate (scat #1052) was 99.75% identical to isolates of *Cryptosporidium* sp. deer mouse genotypes from environmental water (unknown host) (JQ413348) and a deer mouse (KX082684). The isolate from an unknown predator (possibly coyote) (scat #1018) was 100% identical to an isolate of *C. canis* from a coyote (*Canis latrans*) (AY120909) and 99.88% identical to *C. canis* isolates from domestic dogs (*Canis lupus familiaris*), raccoon dog (*Nyctereutes procyonoides*), mink (*Neovison vison*), blue fox (*Vulpes lagopus*), and two isolates of *C. canis* (KT749817-749818) from humans (children). The isolate from a snowshoe hare (*Lepus americanus*) (scat #1296) was 100% identical to *Cryptosporidium* sp. skunk genotype from an Eastern gray squirrel (*Sciurus carolinensis*) (MF411075), and a human (JQ413444), along with sequences from environmental water (EU825736) and storm water (AY737559) (unknown hosts).

Among the three isolates from mountain beavers (*Aplodontia rufa*), the isolate from scat #1162 was 100% identical to *C. ubiquitum* from domestic sheep (*Ovis aries*) (KC608024), roe deer (HQ822139), and laboratory rats (MT102933). The isolate was also 100% identical to twenty-six isolates of *Cryptosporidium* sp. cervine genotype from domestic sheep or goats, one environmental isolate from storm water (AY737592), and humans (AJ849465). The strain from humans (AJ849465) was also *Cryptosporidium* cervine genotype based on accession submissions by Soba et al., 2006, and Trotz-Williams et al., 2006 [29,30]. The other two isolates from scat #1378 and #1379 were only 97.78–97.89% identical to an isolate from storm water (AF262331), 96.79–96.90% identical to an isolate of *Cryptosporidium* sp. skunk genotype (MF411075) from an eastern gray squirrel, and 96.75–96.87% identical to an isolate of *Cryptosporidium* sp. skunk genotype (JQ413444) from humans. This relative lack of sequence similarity to any isolates in GenBank suggests these isolates are likely a novel genotype and are also the first published report of *Cryptosporidium* oocysts from mountain beavers.

Finally, the isolate from western spotted skunk (*Spilogale gracilis*) (scat #1161) was only 98.69‒98.73% identical to strains from storm and environmental water (unknown host) and only 98.33% identical to an isolate from meadow vole (Table 3), which suggests that this isolate from a western spotted skunk was also a novel genotype.

Results of phylogenetic analysis are shown in Figure 1. All *Cryptosporidium* genotypes detected in the wildlife populations in this study clustered in the intestinal *Cryptosporidium* group. Among this cluster, most genotypes grouped with species and genotypes as predicted by the results of the BLAST analysis. For example, the isolate from deer (scat #930) grouped with *C. ryanae*, *C. bovis*, and *C. xiaoi*; the isolate from bobcat (scat #1017) grouped with *C. felis*; the isolate from unknown predator (possibly a coyote) (scat #1018) grouped with *C. canis*; the isolate from a hare (scat #1296) grouped with skunk genotype; the isolate from a mountain beaver (scat #1162) grouped with *C. ubiquitum*; and isolates from other deer mice (scat #R5, R6) grouped with *C. cuniculus*. In contrast, an isolate from a bobcat (scat #1108) and two mountain beavers (scat #1378 and 1379) formed their own cluster and were not closely related to any species or genotypes in GenBank; similarly, the western spotted skunk (scat #1161) and deer mouse (scat #1052) isolates did not directly cluster with other strains, with the closest link being isolates of *C. canis* and an unknown predator (scat #1018).

## 4. Discussion

### 4.1. Apparent Prevalence of Cryptosporidium in Wildlife within Various Watersheds

*Cryptosporidium* species have been widely reported to infect wildlife populations in municipal and rural watersheds in the U.S. and other countries. In Australia, *Cryptosporidium* spp. oocysts were detected in feces from brushtail possum, kangaroo, deer, and rabbit in the Sydney watershed [31]. In another study, *Cryptosporidium* oocysts were detected in 6.7% of eastern grey kangaroo fecal samples in the Sydney hydrological catchment [32]. In a survey of *Cryptosporidium* in animals located in Sydney’s drinking water catchments, *Cryptosporidium* species were detected in fecal samples of 3.6% of kangaroos, 7.0% of cattle, 2.3% of sheep, and 13.2% of rabbits [33]. In a long-term monitoring of *Cryptosporidium* in drinking water catchments in three states across Australia (Western Australia, New South Wales, and Queensland), overall prevalence of *Cryptosporidium* across various host species was 18.3% in a total of 5774 fecal samples from 17 known host species and 7 unknown bird samples from 11 water catchment areas [34]. The prevalence of *Cryptosporidium* spp. oocysts was 1.62% (69/4256) in fecal samples from animals in water catchments supplying the City of Melbourne [35]. In Canada, a lower prevalence of 0.94% for *Cryptosporidium* was found in wildlife, with scat from 6 of 19 wildlife species testing positive at locations along tributaries of the North Saskatchewan River in Alberta [36]. In the U.S., *Cryptosporidium* spp. oocysts were detected in 64% (25/39) of mammal species and 29% (4/14) of bird species of wildlife populations within the Catskill/Delaware watershed of New York City’s water supply system [37] and similarly, in 20.5% species of wildlife from both watersheds (Catskill/Delaware and Croton) of New York City’s water supply system, with most positive animals being mammals [38].

Our study is the first known published report regarding the prevalence of *Cryptosporidium* species in free-ranging wildlife populations in a watershed of a municipal water supply in the Pacific Northwest. The results from this project indicate that the apparent prevalence of *Cryptosporidium* spp. for the different wildlife populations in this protected watershed has a similar range of values to those reported in watersheds in Australia, Canada, and New York State referenced above, with a higher apparent prevalence occurring for one or more species in the Artiodactyla (e.g., Roosevelt elk), Carnivora (e.g., bobcat, cougar, and Western spotted skunk), and Rodentia (e.g., beaver, bushy-tailed wood rat, and mountain beaver) orders. It is important to note that the true prevalence of *Cryptosporidium* in different free-ranging wildlife species in this study could be affected by many factors. These factors include opportunistic versus random sampling, small sample sizes for rare wildlife, different percent recoveries of oocysts from different wildlife scat (e.g., Table 1), and the possibility of resampling individual animals due to sampling of scat and/or unmarked wildlife. Nevertheless, this study provided baseline information of *Cryptosporidium* in wildlife in this Pacific Northwest watershed, which can be used by municipal water agencies to prioritize wildlife monitoring of zoonotic pathogens and to better gauge waterborne health risks from wildlife zoonoses like *C. parvum*.

### 4.2. Genotyping of Cryptosporidium and Zoonotic Species in the Watershed

According to a recent review, species and genotypes of *Cryptosporidium* that are considered zoonotic or potentially zoonotic include (major vertebrate hosts in parenthesis): *C. parvum* (cattle), *C. erinacei* (tree squirrels), *C. scrofarum* (pigs), *C. tyzzeri* (mice), *C. cuniculus* (rabbits), *C. ubiquitum* (cattle), *C. xiaoi* (sheep and goats), *C. fayeri* (kangaroo), *C. bovis* (cattle), *C. suis* (pigs), *C. canis* (dogs), *C. andersoni* (cattle), *C. meleagridis* (turkeys), *C. felis* (cats), *C. muris* (mice), chipmunk genotype I (chipmunks), horse genotype (horses), mink genotype (minks), and skunk genotype (skunks) [5].

Similar to genotyping results from this study, many *Cryptosporidium* species and genotypes with variable zoonotic potential have been reported in wildlife populations in watersheds in the U.S. and other countries. In the Catskill/Delaware watershed of New York City’s water supply system, *Cryptosporidium* genotypes in twelve wildlife species were primarily host-specific, but 38% of sampled wildlife were shedding zoonotic *C. parvum* [39]. In a similar study of *Cryptosporidium* genotypes in wildlife in both watersheds (Catskill/Delaware and Croton) of New York City’s water supply system, most genotypes had no public health significance, but zoonotic *C. parvum*, *C. meleagridis*, *C. muris*, cervine genotype, skunk genotype, chipmunk genotype I, and mink genotype were detected [38]. During a pathogen monitoring program for municipal catchments serving the City of Melbourne, Australia, between 2011 to 2015, at least five species (*C. parvum*, *C. hominis*, *C. cuniculus*, *C. ubiquitum*, and *C. suis*) among thirty-four species and genotypes identified in their animals were likely infectious to humans [35]. Oocyst contamination in water of the South Nation River watershed in Ontario, Canada was associated with both wildlife and livestock *Cryptosporidium* [40]. *Cryptosporidium* contamination in the Wachusett Reservoir, a drinking water source for Boston, Massachusetts, was thought to originate from birds [41], and contamination in the Wissahickon watershed near Philadelphia was thought to originate from deer and geese [42].

PCR amplification of *Cryptosporidium* oocyst DNA from aged scat samples (i.e., not collected per rectum) were hampered in some older samples in this study due to either low concentrations of oocysts, aged and possibly damaged oocysts, and/or the many inhibitors present in feces, as experienced by our laboratory and reported by others [43,44,45]. For example, feces exposed to moderate to high ambient temperature, in part due to exposure to solar radiation, can lead to excystation of oocysts [46,47] and subsequent loss of DNA in the fecal matrix, thereby impeding PCR. In addition, during the course of this study the decision was made to cease attempts at PCR amplification for scat samples with less than 20 to 25 oocysts given the low success rate for these types of samples. Given these challenges and caveats, we attempted to PCR and sequence 70 *Cryptosporidium* positive samples and 12 (17%) of these positive scat samples were successfully genotyped.

The twelve genotyped samples were from six species of hosts, including bobcat, black-tailed deer, deer mouse, snowshoe hare, mountain beaver, and western spotted skunk, and one unknown predator most likely to be a bobcat, cougar, or coyote. Comparison of *Cryptosporidium* genotypes from these wildlife species to sequences in the GenBank by BLAST analysis (Table 3), along with comparison to recognized representative species and zoonotic genotypes by phylogenetic analysis (Figure 1) generated similar inferences with respect to the zoonotic potential of these twelve isolates and where they were positioned within the phylogenetic clusters of related species or genotypes of *Cryptosporidium*, as explained below.

The *Cryptosporidium* sp. isolated from the black-tailed deer was 100% identical to multiple deer genotypes in GenBank, suggesting that this *Cryptosporidium* sp. is host-specific and therefore not likely to be infectious to humans. However, BLAST analysis also indicated that the same sequence was 100% identical to *C. ryanae* from barking deer (Table 3), while phylogenetic analysis also showed higher similarity to sequences representative of *C. ryanae* species (Figure 1). This 100% matching to two differently named species or genotypes of *Cryptosporidium* for a single isolate occurs because the species *C. ryanae* was named in 2008 based on isolates previously identified as deer-like genotype [48]. Given that the *Cryptosporidium* isolate from a black-tailed deer in the current study was 100% identical to a *C. ryanae* isolate in GenBank, these results suggest cross-species transmission of this strain of *Cryptosporidium* could occur between infected black-tailed deer and other common mammalian hosts of *C. ryanae* [48].

*Cryptosporidium* sp. isolated from deer mice in this study exhibited high similarity or were identical (99.75 to 100%) to known *Cryptosporidium* deer mouse genotypes, indicating these isolates were likely host-specific to deer mice with limited potential of cross-host-species transmission and of low zoonotic potential. In contrast, the DNA sequence from a *Cryptosporidium* isolate from a western spotted skunk was only 98.7% similar to *Cryptosporidium* isolates from a meadow vole and storm and environmental waters with unknown animal origin, indicating the *Cryptosporidium* genotype in this infected host is potentially novel with an unknown zoonotic potential.

One bobcat (scat #1017) shed *Cryptosporidium* oocysts that were highly similar (≥99.47%) to *C. felis* from humans and domestic cats, which is a zoonotic species infectious to humans. The *Cryptosporidium* sp. from the unknown predator was 99.88% identical to *C. canis* from a human, dog, and several species of wildlife, and 100% identical to *C. canis* from a coyote. *C. canis* typically infects animals of the Canidae family but is infrequently isolated from infected humans. Based on visual appearance of scat, the suspect host was originally identified as cougar, but the confidence of this identification was low because scat from cougar and coyote can look very similar. If the infected host was indeed a cougar, it is unusual for the species of *Cryptosporidium* to be *C. canis*, unless of course the cougar preyed on an infected coyote. Alternatively, the field personnel may have originally misidentified the scat and instead the actual source was coyote. If this was true, it would not be surprising to detect *C. canis* in a coyote.

Interestingly, the *Cryptosporidium* oocysts from a snowshoe hare were 100% identical to *Cryptosporidium* skunk genotypes from environmental water, humans, and eastern gray squirrels (*Sciurus carolinensis*). Our results indicate this zoonotic skunk genotype may infect a wider range of host species with the potential for cross-species transmission among various wildlife populations in the watershed.

*Cryptosporidium* sp. from a mountain beaver (scat #1162) was 100% identical to *C. ubiquitum* from domestic sheep, roe deer, and laboratory rats, and also 100% identical to numerous isolates of cervine genotype from humans and an environmental isolate. This suggests that this zoonotic isolate from a mountain beaver may be able to infect multiple mammalian species including humans. The original isolates used to establish the species of *C. ubiquitum* were based on the cervine genotype, so it is not surprising that this sequence matches 100% to both *C. ubiquitum* and cervine genotype sequences in GenBank [49].

The other two mountain beaver isolates (scat #1378 and 1379) were only 97–98% similar (or 2–3% dissimilar) to *Cryptosporidium* sp. isolate BB (genotype W6) skunk genotypes from storm water and skunk genotypes from eastern gray squirrels and humans, indicating that this is a novel genotype or a new species of *Cryptosporidium* given its lack of a close sequence match to all archived *Cryptosporidium* samples in GenBank. To our knowledge, this is the first published report of *Cryptosporidium* in a mountain beaver. Based on Figure 1, the closest DNA sequence match to these two new strains of *Cryptosporidium* from mountain beavers was to a bobcat isolate (scat #1108). Either this new strain of *Cryptosporidium* infected both host species, or alternatively, this finding suggests that the bobcat preyed on an infected mountain beaver and subsequently shed oocysts of this new strain of *Cryptosporidium* in its scat. Given that 40% of mountain beaver scat had detectable *Cryptosporidium* oocysts in this study, there was a reasonably good opportunity for predators who consume this large rodent to be exposed to this novel strain of *Cryptosporidium*.

## 5. Conclusions

BLAST and phylogenetic analysis of *Cryptosporidium* in the current study indicated that multiple and diverse species and genotypes of *Cryptosporidium* are present at this location, with some isolates possibly co-circulating within and between wildlife populations in this protected watershed of a major municipal drinking water supply in the Pacific Northwest. Evidence of oocyst exchange between infected prey and their predators was also found. Lastly, a range of *Cryptosporidium* isolates with varying levels of zoonotic potential were identified. Four of the 12 isolates that were speciated or genotyped have a history of association with human infection, but the majority (8/12) of these isolates were not of significant public health concern. Given the small sample size of speciated isolates obtained in this study, we recommend that additional sampling occur in the future to broaden the database regarding which wildlife are infected with which species of *Cryptosporidium* to better understand the public health risks from this protozoan parasite on a municipal watershed.

## Figures and Tables

**Figure 1 microorganisms-08-00914-f001:**
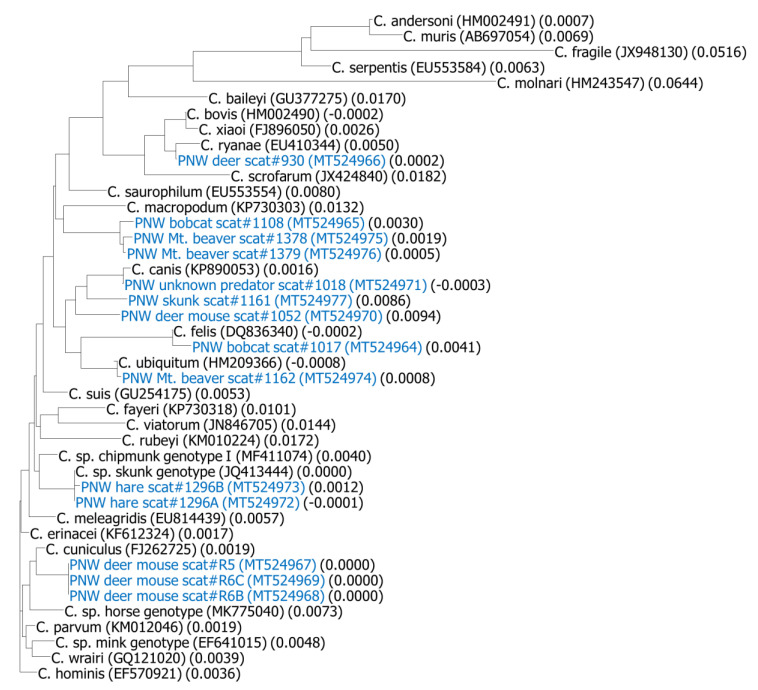
Phylogenetic dendrogram comparing *Cryptosporidium* isolates from a protected watershed in the Pacific Northwest (blue font) and representative archived *Cryptosporidium* species from fish, amphibians, reptiles, birds, and mammals, and selected zoonotic genotypes (black font). GenBank accession numbers are in the brackets behind the names of representative archived species or genotypes of *Cryptosporidium*. Names of isolates from the municipal watershed start with PNW followed by wildlife species, scat #, GenBank accession number, and calculated distance value.

**Table 1 microorganisms-08-00914-t001:** Matrix spike recoveries for *Cryptosporidium* oocysts from scat samples of wildlife.

Wildlife Species	Number of Spiked Scat Samples	Oocyst Recovery (%) Mean (Standard Deviation)
Canada Goose	5	4.0 (1.2)
Snowshoe Hare	5	15.4 (11.5)
Small Rodents	6	6.6 (3.8)
American Beaver	5	15.6 (14.8)
Coyote	6	18.3 (13.9)
Black Bear	5	11.1 (5.4)
River Otter	5	2.2 (2.8)
Cougar	4	31.5 (15.1)
Bobcat	6	16.1 (17.2)
Roosevelt Elk	6	24.8 (14.5)
Black-Tailed Deer	7	15.9 (9.5)

**Table 2 microorganisms-08-00914-t002:** Prevalence of *Cryptosporidium* in wildlife co-located on a protected watershed in the Pacific Northwest, 2013–2016.

Wildlife Species	Dates of Scat Collection	Number of Scat Collected	Number (%) of Scat Positive for *Cryptosporidium*	Number of Positive Samples Successfully Genotyped
**Anseriformes**		**28**	**1 (3.6)**	**0**
Canada goose (*Branta canadensis*)	3/24/2014–4/25/2016	28	1 (3.6)	0
**Artiodactyla**		**99**	**19 (19.2)**	**1**
Roosevelt elk (*Cervus canadensis roosevelti*)	10/4/2013–5/16/2016	38	10 (26.3)	0
Black-tailed deer (*Odocoileus hemionus columbianus*)	10/4/2013–4/18/2016	61	10 (16.4)	1
**Carnivora**		**146**	**31 (21.2)**	**4**
American black bear (*Ursus americanus*)	4/2/2014–4/25/2016	23	0 (0)	0
Bobcat (*Lynx rufus*)	1/6/2014–5/16/2016	45	15 (33.3)	2
Cougar (*Puma concolor*)	5/30/2014–4/25/2016	8	4 (50)	0
Coyote (*Canis latrans*)	1/21/2014–5/10/2016	46	8 (17.4)	0
River otter (*Lontra canadensis*)	10/12/2015–4/12/2016	19	1 (5.3)	0
Western spotted skunk (*Spilogale gracilis*)	6/25/2014–6/26/2014	2	1 (50.0)	1
Unknown carnivore ^a^	5/30/2014	1	1 (100)	1
**Eulipotyphla**		**8**	**0 (0)**	**0**
Trowbridge’s shrew (*Sorex trowbridgii*)	4/29/2014–7/14/2015	7	0 (0)	0
Vagrant shrew (*Sorex vagrans*)	5/7/2014	1	0 (0)	0
**Lagomorpha**		**53**	**5 (9.4)**	**1**
Snowshoe hare (*Lepus americanus*)	10/28/2013–3/9/2016	52	4 (7.7)	1
America pika (*Ochotona princeps*)	7/7/2014	1	1 (100)	0
**Rodentia**		**174**	**14 (8.0)**	**6**
Beaver (*Castor canadensis*)	10/11/2013–1/27/2016	25	3 (12.0)	0
Bushy-tailed woodrat (*Neotoma cinerea*)	6/25/2014–7/15/2015	20	3 (15.0)	0
Deer mouse (*Peromyscus maniculatus*)	2/27/2014–7/16/2015	73	3 (4.1)	3
Douglas squirrel (*Tamiasciurus douglasii*)	8/8/2014	1	0 (0)	0
Long-tailed vole (*Microtus longicaudus*)	7/17/2014	1	0 (0)	0
Mountain beaver (*Aplodontia rufa*)	6/3/2014–7/17/2015	10	4 (40.0)	3
Red tree vole (*Arborimus longicaudus*)	7/22/2014	1	0 (0)	0
Townsend’s chipmunk (*Neotamias townsendii*)	6/10/2014–7/17/2015	42	1 (2.4)	0
Townsend′s vole (*Microtus townsendii*)	7/15/2014	1	0 (0)	0
**Total**		**506**	**70 (13.8)**	**12**

^a^ Likely to be either cougar, coyote, or bobcat.

**Table 3 microorganisms-08-00914-t003:** Comparison of the 18S rRNA gene sequences of *Cryptosporidium* from wildlife co-located on a protected watershed in the Pacific Northwest to sequences of *Cryptosporidium* in the GenBank by BLAST analysis (updated by 12 March, 2020).

Wildlife Species	Scat ID	Sequence Length (bp)	Highly Similar Sequences in the GenBank			
			*Cryptosporidium* Isolates	Hosts or Sources	GenBank Accession no.	Max. Identity (%)
Bobcat	1017	800	*Cryptosporidium felis* isolate 118	Domestic cat	MK886594	99.58
			*Cryptosporidium felis* isolate T7	Domestic cat	MF589920	99.57
			*Cryptosporidium felis* isolate W13866	Humans	HM485433	99.47
	1108	828	*Cryptosporidium* sp. isolate BB (genotype W6)	Storm water	AF262331	97.96
			*Cryptosporidium* environmental sequence isolate 7843-a1 (genotype W18)	Storm water	AY737575	97.65
			*Cryptosporidium* environmental sequence isolate 8278 (genotype W5)	Storm water	AY737594	97.32
			*Cryptosporidium* sp. isolate 1764-Mipe-NA	Meadow vole	KY644567	97.23
Black-tailed deer	930	797	*Cryptosporidium* sp. deer genotype isolates	Sika deer	MN056193-9	100
			*Cryptosporidium ryanae* isolate ZH-07	Barking deer	MK982509	100
			*Cryptosporidium* sp. deer genotype isolate 32	David’s deer	MK571183	100
			*Cryptosporidium* sp. deer genotype isolate 262	Red deer	KX259129	100
			*Cryptosporidium* sp. deer genotype isolate 32	David’s deer	KX259128	100
			*Cryptosporidium* sp. deer genotype isolate 600	Sika deer	KX259127	100
			*Cryptosporidium* sp. deer genotype	White-tailed deer	KR260681	100
			*Cryptosporidium* sp. deer genotype isolate K39_4151	White-tailed deer	KJ867493	100
Deer mouse	R5	830	*Cryptosporidium* sp. ex *Peromyscus maniculatus* isolate 2951	Deer mouse	KX082685	100
	R6-B ^a^	829	*Cryptosporidium* sp. ex *Peromyscus maniculatus* isolate 2951	Deer mouse	KX082685	100
	R6-C^a^	830	*Cryptosporidium* sp. ex *Peromyscus maniculatus* isolate 2951	Deer mouse	KX082685	100
	1052	824	*Cryptosporidium* sp. deer mouse genotype IV (W3) isolate CRY1811	Environmental water	JQ413348	99.75
			*Cryptosporidium* sp. ex *Peromyscus maniculatus* isolate 2828	Deer mouse	KX082684	99.75
Unknown predator	1018	801	*Cryptosporidium canis* isolate 2011	Coyote	AY120909	100
			*Cryptosporidium canis* isolate L25	Domestic dog	MN696800	99.88
			*Cryptosporidium canis* isolate 255	Raccoon dog	MN238765	99.88
			*Cryptosporidium canis* isolate 215	Blue fox	MN238764	99.88
			*Cryptosporidium canis* isolate 28	Mink	MN235856	99.88
			*Cryptosporidium canis* isolates HZ-C2 and C5	Domestic dog	KR999984; KR999987	99.88
			*Cryptosporidium canis* isolate S22 and S25	Humans (children)	KT749818; KT749817	99.88
			*Cryptosporidium canis* isolate	Domestic dog	JN543379; JN543381; JN543383; JN543384;	99.88
			*Cryptosporidium canis* strain CPD1	Domestic dog	AF112576	99.88
			*Cryptosporidium canis*	Domestic dog	AB210854	99.88
Snowshoe hare	1296A ^a^	818	*Cryptosporidium* sp. skunk genotype isolate 14550	Eastern gray squirrel	MF411075	100
			*Cryptosporidium* sp. skunk genotype strain W23573	Humans	JQ413444	100
			*Cryptosporidium* environmental sequence isolate 15081-4 (Skunk genotype (W13))	Environmental water	EU825736	100
			*Cryptosporidium* environmental sequence isolate 8224 (genotype W13) (skunk genotype in Jiang et al., 2005)	Storm water	AY737559	100
	1296B ^a^	829	*Cryptosporidium* sp. skunk genotype isolate 14550	Eastern gray squirrel	MF411075	100
			*Cryptosporidium* sp. skunk genotype strain W23573	Humans	JQ413444	100
			*Cryptosporidium* environmental sequence isolate 15081-4 (skunk genotype W13)	Environmental water	EU825736	100
			*Cryptosporidium* environmental sequence isolate 8224 (genotype W13)	Storm water	AY737559	100
Mountain beaver	1162	831	*Cryptosporidium ubiquitum* isolate A2	Domestic sheep	KC608024	100
			*Cryptosporidium* sp. cervine genotype	Domestic sheep	EU827375	100 ^b^
			*Cryptosporidium* sp. cervine genotype	Domestic goat or sheep	FJ608596-9; FJ608602	100
			*Cryptosporidium* environmental isolate 8056 (genotype W4)	Storm water	AY737592	100
			*Cryptosporidium* sp. SI23 (cervine genotype)	Humans	AJ849465	100
			*Cryptosporidium ubiquitum* isolate P949	Roe deer	HQ822139	100
			*Cryptosporidium ubiquitum* isolate 49	Laboratory rat	MT102933	100
	1378	810	*Cryptosporidium* sp. isolate BB (genotype W6)	Storm water	AF262331	97.78
			*Cryptosporidium* sp. skunk genotype isolate 14550	Eastern gray squirrel	MF411075	96.79
			*Cryptosporidium* sp. skunk genotype strain W23573	Humans	JQ413444	96.75
	1379	807	*Cryptosporidium* sp. isolate BB (genotype W6)	Storm water	AF262331	97.89
			*Cryptosporidium* sp. skunk genotype isolate 14550	Eastern gray squirrel	MF411075	96.90
			*Cryptosporidium* sp. skunk genotype strain W23573	Humans	JQ413444	96.87
Western Spotted skunk	1161	837	*Cryptosporidium* environmental sequence isolate 8057 (genotype W12)	Storm water	AY737558	98.73
			*Cryptosporidium* environmental sequence isolate CRY2984 (genotype W12 variant)	Environmental water	JQ413387	98.72
			*Cryptosporidium* environmental sequence isolate CRY1565 (genotype W12 variant)	Environmental water	JQ178288	98.72
			*Cryptosporidium* sp. (genotype W12)	Storm water	AY007254	98.69
			*Cryptosporidium* sp. isolate 1820-Mipe-NA	Meadow vole	KY644661	98.33

^a^ Two sequences were obtained from the sample.^b^ The sequence was also 100% identical to other 20 isolates of cervine genotype from domestic sheep or goats from the same group of submissions.

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
