# Peer review of "Prevalence and Genotypes of Cryptosporidium in Wildlife Populations Co-Located in a Protected Watershed in the Pacific Northwest, 2013 to 2016"

_microorganisms, 2020, doi:10.3390/microorganisms8060914_

Round 1

Reviewer 1 Report

The manuscript as submitted is over-written. The source of the information was 506 samples (collected by rather casual means) over 2 1/2 years, and processed by poorly described (and therefore presumed to be questionable) procedure. The reason for collection of the samples was "...to allow better assessment Cryptosporidium health risks from human exposure...through drinking water". Of 23 animal species represented, 13 were represented by more than 10 samples and among these 3 were found to be >10% positive (none >15%). More than anything else these results suggest that a more rigorous sampling, sample processing, and analysis procedure would be suggested. 

Given that limited results in light of the nature of the objective a reasonable conclusion might have been that further work was needed. Given the limited data the description of the genotyping is far more extensive than justified.

See additional comments in the manuscript, attached.

Reviewer 2 Report

The manuscript presented for review (microorganisms-818504-peer-review-v1) concerns the epidemiology of Cryptosporidum spp. and the identification of natural wildlife, that serve as reservoir of its  strains responsible for human infections, in a protected watershed being a source of municipal water. Parasites of the genus Cryptosporidium are ubiquitous zoonotic pathogens of humans and animals and are responsible for significant number of water-borne outbreaks worldwide. Although farm and domestic animals are usually a reservoir for it, wild animals can also be its vector.

In presented paper, among identified strains of Cryptosporidium only few were of public health concern. However, the Authors pointed out that monitoring of this reservoir of municipal water for strains, that cause human infections is necessary, due to the identification such strains and genotypes of unknown pathogenicity in wildelife scat samples.

The general remark:

Before publication process the Authors should obtain GenBank accession no. for sequences from new isolates. Did Authors make submission of sequences of new isolates to GenBank? In Figure 1 showing the phylogenetic dendrogram only accession numbers for reference sequences can be seen.

In addition, the current dendrogram is illegible. Please change the font to make it transparent and legible. In the signature, please add sequence based on which the dendrogram has been generated. This information is contained in the materials and methods chapter, but the drawing should be an autonomous source of information and should not require access to such data elsewhere.

Reviewer 3 Report

Li and colleagues in their manuscript presented very interesting data on the genetic diversity of Cryptosporidium in the wild animals in a protected watershed in the Pacific Northwest. Since the Cryptosporidium is an emerging threat for human health, the prevalence of this pathogen in animals (wild and domestic), as well as the transmission between different hosts are the necessary information to develop efficient antimicrobial strategy.

The results are clear and the paper is well written, and it was a pleasure to read the manuscript. I have only few minor comments:

  1. Figure 1 quality could be better. It is quite hard to read it. And maybe it would be good to highlight the isolates from the watershed area.
  2. Authors analysed the Cryptosporidium genetic diversity in the collected faeces. It would be extremely interesting to analyse the Cryptosporidium in the water from the analysed area. Did you try to isolate the Cryptosporidium oocysts and analyse the genetic diversity?

Round 2

Reviewer 1 Report

I hate it when I get review comments on my own submittals like the ones I have made in the first review...and here...so I apologise. The following are the issues that motivate the comments:

  • Only a few published reports of this type...particularly for USA mountain watersheds...exist--This suggesting that when (if?) published the results will influence other work and possibly public policy, thus, details and implications will almost inevitably be taken at face value;
  • From world-wide published work it should be clear that virtually all terrestrial mammals (perhaps not each individual) should be considered routinely...that is assumed to be...carriers of Cryptosporidium (as well as other potentially waterborne pathogens). Logic supports the inference that all Cryptosporidium emanating from a watershed like the one studied must be considered as potentially of a human infectious type;
  • The samples in this study cover a range of some 20 animal species...around 50 animal species have been catalogued in Western Cascade forests. No indication of the fecal-weighted importance, species resident population numbers, or habitat preference (e.g. riparian, sylvan...) is mentioned. Perhaps this is irrelevant with respect to the finding that some of the typed Cryptosporidium are (may be) of a human infective type
  • Other picky features of sampling could be raised: 
    • At what time of year were the individual samples (limit to the 70 C-positive samples?)
    • Did sampling include snow covered areas and times of year?
    • Where were the sampling locations? (PWB water quality management information provides some additional description but not much detail)
    • Very specific animal species are identified as the source of individual samples...However, for scat collected as described perhaps simply coyote, bear, deer, rabbit...etc would have been sufficient...Do the species names imply a greater degree of knowledge than is verifiable...or even useful?;

         The point of this is with respect to information that a reader of the paper will glean...and infer in the future. What is the real and useful scientific contribution of the data encompassed?

  • Reading and reflecting on the Abstract, and knowing that many (rather lazy) authors of journal articles read and reflect what is in an abstract, the sentence citing the percentage of positive samples by animal species seems to suggest greater importance than the numbers of cougar & bobcat...that are not riparian or aquatic...might warrant emphasis...or that the 3/25 beaver might be less important than the 4/10 mountain beaver...or perhaps 10/38 elk...known to favour water and maybe seen defacating while wading? 

Overall, the manuscript is well-written and includes useful new information. Perhaps the authors can read the above comments and understand them in the spirit in which they are written. To spend time on reviewing any manuscript intended to add to a body of knowledge without paying attention to details would be time wasted. The authors must choose what they wish to convey to any ultimate reader. This reviewer will not require any revision.

Author Response

Thank you for your comments and your request for no further revisions.

Based on your comment about inferences made from the abstract regarding high or low fecal prevalences of Cryptosporidium oocysts in different wildlife species, we revised sentences 16 to 18 to read, "Prevalence of Cryptosporidium varied among species of wildlife, with higher prevalences observed in cougars (50.0%), mountain beavers (40.0%), and bobcats (33.3%), but none of these species are riparian-dependent".

The remainder of the reviewer's narrative did not request additional revisions.